# Anti-Gr-1 Antibody Provides Short-Term Depletion of MDSC in Lymphodepleted Mice with Active-Specific Melanoma Therapy

**DOI:** 10.3390/vaccines10040560

**Published:** 2022-04-04

**Authors:** Peter Rose, Natasja K. van den Engel, Julia R. Kovács, Rudolf A. Hatz, Louis Boon, Hauke Winter

**Affiliations:** 1Department of Cardiac Surgery, University Hospital Heidelberg, 69120 Heidelberg, Germany; 2Translational Lung Research Center (TLRC), Member of German Center for Lung Research (DZL), 69120 Heidelberg, Germany; engel71@live.nl (N.K.v.d.E.); julia.kovacs@med.uni-muenchen.de (J.R.K.); rudolf.hatz@med.uni-muenchen.de (R.A.H.); hauke.winter@med.uni-heidelberg.de (H.W.); 3Department of Thoracic Surgery, University Hospital Munich, 81377 Munich, Germany; 4Polpharma Biologics, 3584 CM Utrecht, The Netherlands; louis.boon@polpharmabiologics.com; 5Department of Surgery, Thoraxklinik at University Hospital Heidelberg, 69126 Heidelberg, Germany

**Keywords:** immunotherapy, vaccination, myeloid-derived suppressor cells, melanoma, T-Lymphocytes

## Abstract

Lymphodepletion, reconstitution and active-specific tumor cell vaccination (LRAST) enhances the induction of tumor-specific T cells in a murine melanoma model. Myeloid-derived suppressor cells (MDSC) may counteract the induction of tumor-reactive T cells and their therapeutic efficacy. Thus, the aim of the study was to evaluate a possible benefit of MDSC depletion using anti-Gr-1 antibodies (Ab) in combination with LRAST. Female C57BL/6 mice with 3 days established subcutaneous (s.c.) D5 melanoma were lymphodepleted with cyclophosphamide and reconstituted with naive splenocytes. Vaccination was performed with irradiated syngeneic mGM-CSF-secreting D5G6 melanoma cells. MDSC depletion was performed using anti-Gr-1 Ab (clone RB6-8C5). Induction of tumor-specific T cells derived from tumor vaccine draining lymph nodes (TVDLN) was evaluated by the amount of tumor-specific interferon (IFN)-γ release. LRAST combined with anti-Gr-1 mAb administration enhanced the induction of tumor-specific T cells in TVDLN capable of releasing IFN-γ in a tumor-specific manner. Additional anti-Gr-1 mAb administration in LRAST-treated mice delayed growth of D5 melanomas by two weeks. Furthermore, we elucidate the impact of anti-Gr-1-depleting antibodies on the memory T cell compartment. Our data indicate that standard of care treatment regimens against cancer can be improved by implementing agents, e.g., depleting antibodies, which target and eliminate MDSC.

## 1. Introduction

Immunotherapy has long found its way into melanoma treatment. Recently, immune checkpoint-inhibition of CTLA-4 and PD-1 has emerged as the frontline option for the treatment of patients with advanced stages of the disease, targeting negative regulations of immune responses and thereby activating tumor-specific cytotoxic T cells [1,2,3]. In contrast, active-specific immunotherapy engages autologous tumor cells as a whole cell vaccine to provide a variety of potential tumor antigens [4]. In many tumors, these tumor antigens have not been identified and often represent weak self-antigens, unlikely to induce a robust anti-tumor T cell immune response [5].

Conducting active-specific tumor vaccination in lymphopenic hosts has provided promising results in murine gastric and melanoma models, as well as in mice challenged with cells derived from spontaneously occurring mammary tumors of neu transgenic mice [5,6,7]. 

This effect might partially result from a lymphopenia-induced proliferative stimulus on pre-existing and newly established tumor-directed T cells, especially when homeostatic proliferation coincides with antigen encounter [7,8,9,10]. Preclinical studies with active-specific and adoptive immunotherapy in a murine melanoma tumor model demonstrated that this strategy enhances the induction of tumor-specific T cells, improving therapeutic efficacy [5,11]. Although van den Engel et al. and others could show an enhanced anti-tumor immune response using a whole cell vaccine combined with GM-CSF exposure following lymphodepletion [5,6,7], generating tumor-directed T cells does not necessarily translate into an effective trafficking to the tumor site and tumor cell-killing [12]. A major obstacle is a locally immunosuppressive tumor environment of which MDSC represent a major component [13].

MDSC represent a heterogeneous population of immature myeloid cells and are known for their immunosuppressive properties in the tumor microenvironment, facilitating tumor growth and metastasis [14,15]. Their contribution to an impaired efficacy of immunotherapy has been shown for adoptive cell therapy, dendritic cell (DC) vaccination and checkpoint inhibition [16]. MDSC expansion occurs in pathological conditions, including malignant disease, upon alteration in myelopoiesis followed by accumulation in peripheral lymphoid organs as well as in the tumor microenvironment [13]. In mice, MDSC are commonly defined by the co-expression of the myeloid delineation markers CD11b and Gr-1. Based on the expression of the cell surface markers Ly6G and Ly6C, the two major subtypes of MDSC with distinct phenotype, morphology, and immunosuppressive properties can be distinguished: CD11b^+^ Ly6G^−^ Ly6C^high^ monocytic (M-MDSC) and CD11b^+^ Ly6G^+^ Ly6C^+^ polymorphonuclear (PMN-) MDSC [13,17]. In the periphery, PMN-MDSC represent the largest portion of MDSC, when upregulated under different pathological conditions [13,18]. In the tumor environment, MDSC become more suppressive and M-MDSC outweigh PMN-MDSC in frequency [13,18,19]. The immunosuppressive capacities of MDSC are mediated in an antigen-specific and non-specific manner [14]. In lymphoid organs of the periphery MDSC mainly act antigen-specific with mechanisms including NO and ROS, nutritional deprivation of T cells from L-arginine, L-tryptophan and L-cysteine, impairment of T cell homing and production of TGF-β and IL-10, thus creating an immunosuppressive milieu. Within the tumor environment, MDSC act in a more non-specific way, including upregulation of ARG-1 and iNOS and the inhibitory surface-expression of PD-L1 [13,20]. Regulatory T cells (Treg), a sub-population of CD4^+^ T cells capable of downregulating anti-tumor immune responses, are induced by MDSC in the periphery and are attracted to the tumor site by secretion of the chemokines CCL4 and CCL5 [13,21,22]. 

Different efforts have been made to modulate or even ablate MDSC aiming at improving the outcome of malignant disease in mice and men [23]. There is evidence that depletion of MDSC, e.g., by applying triperpenoids, all-trans-retinoic acids or nitroaspirin, may improve therapeutic efficacy [24]. Antibody-mediated MDSC depletion using anti-Gr-1 or anti-Ly6G monoclonal antibodies (mAb) led to a complete but temporary organ-dependent depletion of MDSC [25,26,27,28]. Long-term treatment of mice with an anti-GR1 antibody combined with a therapeutic BMA-OVA vaccination induced a pronounced tumor reduction or even a complete tumor eradication in a murine lung tumor model, as reported by Srivastava et al. [27].

Thus, a promising approach to improve the therapeutic efficacy of anti-tumor vaccination is the combination of different immunotherapeutic strategies to avoid the threatening possibility of tumor immune escape. 

Different leukocyte populations are involved in anti-tumor reactions. Especially, high frequencies of CD8^+^ cytotoxic and memory T cells were found to be associated with better outcomes in most human cancers [29,30,31]. Effector CD8^+^ T cells (T_eff_) arise from naive T cells upon antigen presentation and co-stimulation by antigen-presenting cells. This fundamental process in anti-tumor immunity mainly occurs in TVDLN but may also happen in direct contact with the tumor. T_eff_ yield robust anti-tumor responses, but lack in durability and long-term activity [32]. This in turn is the main characteristic of memory CD8^+^ T cells, which remain present following first line responses [32,33]. Furthermore, memory CD8^+^ T cells, in contrast to their effector counterparts, seem to be superior in their cytotoxic abilities [34,35,36]. Both central-memory (T_cm_) and effector-memory (T_em_) T cells contribute to an anti-tumor response, where T_em_ rapidly acquire effector function, present with enhanced killing capacity and largely contribute to tumor-specific IFN-γ production [32]; T_cm_ in contrast reside in secondary lymphatic tissues due to their expression of CD62L and CCR7 and are characterized by a less pronounced effector status and a more stem-cell-phenotype with strong proliferative capacities, giving rise to T_em_ (and T_eff_) upon antigen restimulation [37,38]. Overall, compared to T_em_, T_cm_ were shown to have a more prominent role in anti-tumor responses [33,37,39].

Here, we investigated the therapeutic efficacy of a treatment that combines active-specific tumor-cell vaccination with mGM-CSF-secreting D5 melanoma cells and an antibody-mediated depletion of MDSC in a model of murine D5-melanoma. The aim of the study was to evaluate the frequencies of MDSC and CD8^+^ T cells following LRAST treatment and to elucidate whether depletion of MDSC would provide a benefit in anti-tumor treatment.

## 2. Materials and Methods

### 2.1. Mouse Strains and Cell Lines

Wild-type C57BL/6 mice (WT, Ptprcb = CD45.2^+^, Charles River Laboratories International, Inc., Sulzfeld, Germany) between 8–12 weeks of age were used for in vivo experiments, as well as for the generation of single-cell suspensions from lymphatic nodules (LN) and spleen. Congenic B6.SJL-Ptprca Pepcb/BoyCrl mice (CD45.1^+^; Charles River Laboratories International, Inc., Calco, Italy) served as spleen cell donors for reconstitution of wild-type mice after lymphodepletion. Mice were kept under standard pathogen-free conditions in the animal facility of the Walter-Brendel Center, Ludwig-Maximilian-University, Munich. The animal experiments were performed after approval by the local regulatory agency (Regierung von Oberbayern, Munich, Germany, Az. 55.2-1-54-2532-184-12/55.2-1-54-2532.8-42-13). B16BL6-D5 (D5) is a poorly immunogenic subclone of B16BL6 melanoma [40]. D5-G6 is a clone of D5 that was stably transduced with a murine GM-CSF retroviral MFG vector (provided by Dr. M. Arca, University of Michigan, Ann Arbor, MI, USA). D5-G6 cells secrete approximately 200 ng/mL/10^6^ cells/24 h GM-CSF [41]. The MCA 310 fibrosarcoma and LLC1 lung carcinoma cell lines were kindly provided by Dr. B.A. Fox (Portland, OR, USA). 

### 2.2. Media and Reagents

For cell culture of MCA310-cells RPMI (Roswell Park Memorial Institute) 1640 medium was used, supplemented with 10% fetal calf serum (FCS “Gold”; PAA Laboratories, Cölbe, Germany), 2 mM sodium pyruvate (Lonza, BioWhittaker, Walkersville, MD, USA), 0.2 mM non-essential amino acids (Lonza, BioWhittaker), 4 mM L-glutamine and 50 µM β-Mercaptoethanol (Sigma-Aldrich, St. Louis, MO, USA). D5-, D5G6- and LLC1-cells required DMEM (Dulbecco’s Modified Eagle Medium), containing 10 % FBS, 1 mM sodium pyruvate, 0.1 mM non-essential amino acids and 4 mM L-glutamine.

### 2.3. Preparation of Single Cell Suspensions

Mice were killed by cervical dislocation after inhalation of anesthesia with isoflurane (Forene 100% *v*/*v*, Abbott GmbH & Co.KG, Wiesbaden, Germany). Spleen, axillary and inguinal LN (in the following called tumor vaccine—draining LN “TVDLN”, in the case when vaccination has been performed), and tumors were removed under sterile conditions. Organs were disrupted using cannulas and syringe-stomps. Single cells were resuspended with FBS (1%) containing PBS and filtered over a 100 µm cell-filter.

### 2.4. In Vivo Treatment of Mice (LRAST)

For the in vivo experiments, lymphodepletion was induced by intraperitoneal (i.p.) injection of 200 mg/kg cyclophosphamide 3 days after tumor inoculation, followed by i.v. reconstitution with 20 × 10^6^ naive CD45.1^+^ spleen cells and active-specific tumor cell vaccination using 5–10 × 10^6^ irradiated mGM-CSF-secreting D5G6 cells. The cyclophosphamide dose was chosen since earlier studies had shown an increased proliferation and long-term survival of antigen-specific T cells at this particular dose, alone or in combination with fludarabine [18,27]. Naive, non-lymphopenic mice served as control. Tumor development was followed by serial measurements of the tumor diameters and was depicted as tumor size (mm^2^) = d × D, where d and D were the shortest and the longest tumor diameter, respectively.

### 2.5. Depletion of MDSC

Starting on the day of lymphodepletion (day 0), 230 µg anti-Gr-1 monoclonal antibodies (clone RB6-8C5) were i.p. injected every other day until mice were euthanized. Anti-phytochromatin antibodies (isotype control) served as control.

### 2.6. In Vitro T Cell Activation and Expansion

For T cell analyses, mice were vaccinated by s.c. injection with 2.0 × 10^7^ irradiated D5G6 tumor cells on four sites near the extremities (5.0 × 10^6^ per injection). Where indicated, lymphodepletion and reconstitution were performed as described above. TVDLNs were harvested nine days after vaccination, and lymph node cells were polyclonally activated with an anti-CD3 monoclonal antibody (mAb; 5 μg/mL, 2C11, kindly provided by Dr. H.M. Hu, Portland, OR) for 2 days at 4.0 × 10^6^ cells/mL in complete medium (CM) in 24—well plates. Subsequently, cells were supplemented with 60 IU/mL of interleukin-2 (IL-2, Proleukin, Chiron, Ratingen, Germany) for 4 days. After 4 days, cytokine release assays were performed as described elsewhere with the following modifications [42]: TVDLN cells (10^6^ cells) were washed and cultured alone or stimulated with tumor cells (0.2 × 10^6^ cells), or immobilized anti-CD3 antibody in 1 mL of CM supplemented with gentamycin (Lonza, Cologne, Germany) and 60 IU IL-2/mL in a 48-well tissue culture plate at 37 °C, 5% CO_2_ for 18 h. The tumor targets included the tumor cell line used for vaccination (D5). LLC1 and MCA310 tumor cells served as negative controls. Supernatants were analyzed by ELISA. 

### 2.7. ELISA

IFN-γ was measured in supernatants by conventional sandwich ELISA, using mAb AN-18 and biotinylated mAb R4-6A2 (BD Biosciences, Heidelberg, Germany). Supernatants were analyzed in duplicate. Extinction was analyzed at 405/490 nm on a Spectra Fluor microplate ELISA reader (TECAN, Crailsheim, Germany) with the EasyWin software (TECAN). The detection limit of the ELISA for IFN-γ was 125 pg/mL.

### 2.8. Flow Cytometry

For surface staining, cells were washed with PBS and suspended in PBS supplemented with 0.5% (*w*/*v*) bovine serum albumin (BSA) and 0.02% (*w*/*v*) sodium azide. Non-specific binding of antibodies to Fc receptors was blocked by preincubation of the cells with rat anti-mouse CD16/CD32 monoclonal antibodies 2.4G2 (1 μg/10^6^ cells, BD Biosciences) for 15 min. Subsequently, the cells were incubated with the mAb of interest for 30 min at 4 °C, washed and analyzed using a FACS Calibur or a LSRII (BD Biosciences). Dead cells were excluded using Zombie Yellow staining. Data were analyzed using the FACS-Diva software (Version 4.0.2) and FlowJo^®^ software (Version 10.5.0). The following reagents and mAbs against murine antigens were used: allophycocyanin (APC)-conjugated anti-mouse/human CD44 (eBioscience), anti-mouse CD8a (Biolegend, including all following), Ly6C; allophycocyanin/Cy7 (APC/Cy7)-conjugated anti-mouse CD3e, Ly6G; fluorescein isothiocyanate (FITC)-conjugated anti-mouse CD45.2; Pacific Blue-conjugated anti-mouse MHC-II; phycoerythrin (PE)-conjugated anti-mouse CD11b, CD3e, CD62L, Ly6G, goat anti-rat IgG; phycoerythrin/Cy7 (PE/Cy7)-conjugated CD8a, CD11b; Peridinin-chlorophyll proteins/Cy5.5 (PerCP/Cy5.5)-conjugated anti-mouse CD45.1. 

### 2.9. Statistical Analysis

Statistical analysis was performed using GraphPad Prism 8 (GraphPad Software Inc., La Jolla, CA 92037, USA). Two-way ANOVA or unpaired Student’s *t* test was used as indicated. Values of *p* < 0.05 were considered to be statistically significant (* *p* < 0.05; ** *p* < 0.01; *** *p* < 0.001; ns *p* > 0.05).

## 3. Results

### 3.1. MDSC Subsets Temporarily Accumulate in Spleen and Peripheral Blood after LRAST

The frequencies of CD11b^+^ Ly6G^+^ Ly6C^+^ PMN-MDSC and CD11b^+^ Ly6G^−^ Ly6C^high^ M-MDSC were assessed in the peripheral blood and the spleen of tumor-bearing wild-type C57BL/6 mice, at indicated time points following lymphodepletion (day 0) and active-specific tumor-cell vaccination (day 1). FACS analyses were performed starting on the day of lymphodepletion (peripheral blood) or 3 days later (spleen; Figure 1). 

About 7 and 10 days after treatment initiation (LRAST) CD11b^+^ MHCII^−^ myeloid cells originating in the recipient mouse (CD45.2^+^) were significantly increased in the spleen and the peripheral blood (Figure 1a,c). We observed a significant increase in CD45.2^+^ PMN-MDSC in the blood (Figure 1c), a remarkable but not significant increase for cells resembling PMN-MDSC in the spleen (Figure 1a), and M-MDSC in blood and spleen (Figure 1a,c). CD45.2^+^ PMN-MDSC accounted for 51% of total CD45.2^+^ leukocytes in the peripheral blood of tumor-bearing mice, 7 days after LRAST. In contrast, reconstituted CD45.1^+^ PMN- or M-MDSC from donor mice were not found in a considerable extent in both organs and did not show similar kinetics compared to their CD45.2^+^ counterparts over time (Figure 1b,d). One month following LRAST, frequencies of CD45.2^+^ PMN- and M-MDSC were back to values comparable to baseline at day 0.

Thus, we hypothesized that the pronounced occurrence of potentially immunosuppressive MDSC following LRAST treatment might be a key limiting factor for treatment efficacy.

### 3.2. RB6-8C5 Eliminates MDSC in the Initial Treatment-Phase

As we had hypothesized that MDSC inhibit the induction of tumor-specific T cells, we injected MDSC-depleting antibodies i.p. in combination with LRAST treatment. Dose and dosing interval were chosen according to previous reports regarding antibody-mediated depletion of MDSC for therapeutic and non-therapeutic purposes and in combination with other immunotherapeutic approaches [25,26,27,43,44,45,46]. To deplete MDSC, we injected 230 µg anti-Gr-1 mAb i.p. every other day, starting on the day of cyclophosphamide administration (Figure 2a). As a control, mice received an equivalent dose of isotype control following the same time schedule. Efficacy of MDSC depletion using anti-Gr-1 Ab was monitored in peripheral blood in short time intervals 24 or 48 h after the last antibody administration and up to 24 days after treatment initiation (Figure 3c,d). 

Due to an identical binding site, staining of Ly6G epitopes with anti-Ly6G (1A8) or anti-Gr-1 fluorochrome-conjugated antibodies is restricted when Gr-1 antibodies (RB6-8C5) are being used. Therefore, to reveal hidden PMN-MDSC and to elucidate the proportion of PMN- and M-MDSC with cell-bound RB6-8C5, we used a secondary antibody (“2nd Ab”) approach. Peripheral leukocytes from RB6-8C5-treated mice were obtained and stained with PE-conjugated goat anti-rat IgG. Corresponding plots (Figure 2b) indicate the portion of PMN- and M-MDSC with cell-bound RB6-8C5. Then, 48 h (day 2) and 96 h (day 4) following first anti-Gr-1 mAb administration, a significant depletion of PMN-MDSC was observed (Figure 2b,c). Secondary antibody staining revealed no relevant amount of cell-bound RB6-8C5 in the PMN-gate on both days. M-MDSC frequencies in mice following MDSC depletion appeared to be lower, compared to values in mice without RB6-8C5, but failed to be significantly reduced. A portion of the cells located in the M-MDSC gate was bound by RB6-8C5 on days 2 and 4 (0.6% on day 2; 1.6% on day 4). 

Thus, anti-Gr-1 mAbs seem capable of sufficiently depleting circulating PMN-MDSC, while M-MDSC appear to be altered in frequencies, but not depleted.

### 3.3. PMN-MDSC Recur Despite Long-Term Treatment with RB6-8C5

In our experiments, the MDSC-depleting antibody was administered i.p. every other day for approximately 4 weeks. Since both MDSC subsets express Ly6C at the surface and anti-Ly6C fluorochrome-conjugated antibodies do not interfere with Rb6-8C5, we plotted CD11b^+^ Ly6C^+^ cells to depict all MDSC, which exhibit secondary antibody binding over time (Figure 3a), representing the percentage of anti-Gr-1 mAb bound MDSC. Compared to the control, which showed absence of RB6-8C5-binding, the CD11b^+^ Ly6C^+^ cells, which represent both MDSC subsets, exhibited increasing RB6-8C5-binding over time, with 0.0% on day 0 (before RB6-8C5 administration), 8.9% on day 2, 62.8% on day 4, and 64.6% on day 7 after the first dose of anti-Gr-1 mAb.

RB6-8C5-bound CD11b^+^ MHCII^−^ myeloid cells revealed a dynamic shift within the Ly6G/Ly6C-plot (Figure 3b). Red dots, indicating RB6-8C5-bound cells, appear in the Ly6G positive and negative region of the Ly6G/Ly6C-plot, but exhibit equal Ly6C-fluorescence intensity, when comparing different mice of the same treatment group at day 7 (Figure 3b). As the significance of antibody-bound MDSC occurring in respective MDSC-gates is not yet clear, we compared adjusted frequencies of PMN- and M-MDSC (all MDSC in the PMN- and M-MDSC gate minus RB6-8C5-bound cells), as well as overall cell frequencies of respective gates (PMN- and M-MDSC including RB6-8C5-bound cells) from LRAST + RB6-8C5-treated mice to MDSC from mice treated only with LRAST (Figure 3c,d). In this manner we tried to address whether RB6-8C5 can induce long-term MDSC depletion.

Depleting antibody was evident on the surfaces of PMN- and M-MDSC up to day 15 after onset of i.p. antibody administration. The portion of PMN-MDSC net of RB6-8C5-bound cells was significantly reduced on days 7 and 15. Remarkably, when the maximum increase in PMN-MDSC after LRAST could be expected (day 7, Figure 1c), PMN-MDSC, even including the RB6-8C5-bound portion, were significantly reduced (Figure 3c). Nevertheless, on both days (7 and 15), a considerable percentage of PMN-MDSC (less pronounced for M-MDSC) showed RB6-8C5-binding, as indicated by secondary antibody staining (Figure 3c). Trends for frequency kinetics for M-MDSC appeared to be different from those of PMN-MDSC, with an initial increase seen on day 7, followed by a reduction by day 15 (Figure 3d). At the end of the experiment (day 24), both PMN- and M-MDSC frequencies where back to baseline in mice treated with LRAST + RB6-8C5 as compared to the control group. 

These data demonstrate that a successful depletion of MDSC, especially PMN-MDSC, can be achieved for a short period of time, despite continuous administration of depleting antibodies. Further, these long-term results indicate a different impact of anti-Gr-1 mAbs on M-MDSC compared to PMN-MDSC, since no significant depleting effect on M-MDSC and little binding of RB6-8C5 could be observed within 24 days.

### 3.4. MDSC Depletion Improves Vaccination Responses

In order to evaluate the effect of MDSC depletion on the induction of tumor-specific T cells, IFN-γ release from TVDLN of LRAST-treated mice was assessed. Therefore, TVDLN were isolated nine days following LRAST and analyzed in a cytokine release assay. Cytokine responses were evaluated upon restimulation with D5-melanoma cells or control tumor-cell lines (LLC1 and MCA310). Following LRAST treatment alone, an increased D5-melanoma-specific IFN-γ release from TVDLN was observed (577 pg/mL; Figure 4a). While the administration of MDSC-depleting antibodies (RB6-8C5) alone did not have any detectable effect on cytokine release with any tumor cell line, we observed a remarkable increase in the D5 tumor-specific IFN-γ secretion from TVDLN in mice treated with the combination of LRAST and RB6-8C5 mAb (8193 pg/mL; Figure 4a).

Tumor growth was recorded starting 7 days after onset of the treatment. Compared to the control group both, LRAST and LRAST + RB6-8C5 led to a delay in tumor progression (Figure 4b). Within the first two weeks after treatment initiation, tumor sizes of animals treated with the combination of LRAST and MDSC depletion were significantly smaller than with LRAST alone (Figure 4b).

### 3.5. Use of RB6-8C5 Leads to Alteration of Memory CD8^+^ T Cells

Effects of a lymphodepleting preconditioning with cyclophosphamid and the administration of antibodies targeting Gr-1 epitopes on CD8^+^ cells and subsets thereof have already been investigated and will be discussed below [11,47,48]. In the treatment groups containing lymphodepletion with cyclophosphamide, frequencies of CD8^+^ cells increased during recovery from cyclophosphamide, followed by a reduction below the baseline and the values of control groups (RB6-8C5 alone and Isotype, Figure 5a). RB6-8C5 treatment had no significant impact on the frequency of CD8^+^ cells compared to untreated mice (Isotype group, Figure 5a). Regarding the administration of MDSC-depleting antibodies (anti-Gr-1 mAb, RB6-8C5) Matsuzaki et al. showed that the Gr-1 epitope is not only attributed to neutrophils and a subset of mouse DC but is also expressed on memory type CD8^+^ T cells [48]. In the present investigation, we confirmed the expression of the Gr-1-epitope on CD8^+^ T cells, but not on CD4^+^ T cells of female C57BL/6 mice (Figure 5b and data not shown). The use of RB6-8C5 for MDSC depletion as described above also led to a reduction in frequencies of CD8^+^ CD44^high^ CD62L^+^ “central” and CD8^+^ CD44^high^ CD62L^−^ “effector” memory cells (Figure 5c). To address to which extent Gr-1^+^ CD8^+^ cells are affected by RB6-8C5 administration, we stained peripheral CD45.2^+^ blood cells for the indicated markers. In groups that included MDSC depletion consistently, lower proportions of Gr-1^+^ CD8^+^ central and effector memory cells were observed compared to the control group. To evaluate whether CD8^+^ memory cells were depleted and not simply masked by cell-bound RB6-8C5, we assessed their percentage of peripheral leukocytes. While CD8^+^ central memory cells were consistently reduced in groups with RB6-8C5 administration, the portion of CD8^+^ memory effectors in mice with LRAST + RB6-8C5 treatment steadily increased over 7 days following an initial decline after initiation of MDSC depletion. This resulted in an increasing T_em_:T_cm_-ratio over time (Figure 5d). On day 7, frequencies of CD8^+^ effector memory cells from mice treated with LRAST + RB6-8C5 surpassed both control groups. Similar cell kinetics were not found in mice treated with lymphodepletion but without vaccination, followed by MDSC depletion (LR + RB6-8C5). Thus, we conclude that administration of a whole cell vaccine after lymphodepleting preconditioning drives the expansion of CD8^+^ memory cells with an effector phenotype (CD62L downregulation).

## 4. Discussion

The recent success of immune checkpoint inhibition in the therapy of malignant melanoma and various other cancer entities emphasizes the significance of tumor-reactive cells, especially effector T cells [32,49,50,51]. However, tumors escape immune surveillance acquiring different accesses, including reduction of immune recognition and immune activation, developing resistance to immune effector mechanisms and establishing an immunosuppressive tumor microenvironment [52]. Focusing on melanoma, with a high risk of disease recurrence in thicker, nodular or mucosal forms, new systemic treatments are necessary for the management of this condition [53]. MDSC were shown to be key mediators of immunosuppression in the tumor microenvironment, facilitating tumor outgrowth, metastasis, and negatively influencing the efficacy of immunotherapy of cancer [54,55,56,57,58,59,60].

In our experiments using a murine melanoma model, we applied a combined immunotherapeutic approach, LRAST, consisting of lymphodepletion with the alkylating agent cyclophosphamide, followed by i.v. reconstitution with naive congenic spleen cells and active-specific tumor vaccination using GM-CSF-secreting whole tumor cells. In the poorly immunogenic D5-melanoma model, we intended to improve T cell immunization at different levels simultaneously [61]. Induction of a lymphopenic environment should empower T cells with a homeostatic drive to proliferate and, by simultaneous exposure to tumor-antigens via whole cell vaccine, ensure that preferentially tumor-directed T cells colonize empty lymphatic niches [62]. GM-CSF is a hematopoietic cytokine and is often used as an adjuvant in immunotherapeutic regimes, especially vaccination strategies [63,64,65,66]. It acts to promote the local recruitment of antigen-presenting cells and improves their maturation, thus enhancing antigen presentation to T cells in TVDLN [67].

In line with previous reports [7,62], we observed a remarkable increase in frequencies of CD11b^+^ Ly6C^high^ Ly6G^−^ (phenotype of M-MDSC) and CD11b^+^ Ly6C^+^ Ly6G^+^ (phenotype of PMN-MDSC) cells (together also attributable as CD11b^+^ Gr-1^+^ cells) following LRAST. Within 24 days after lymphodepletion with 4.0 mg i.p. cylophosphamide per animal, PMN- as well as M-MDSC increased, peaking at day 7 in peripheral blood and day 10 in the spleen. Similar cell kinetics were reported by Salem et al. in blood, spleen and bone marrow of C57BL/6 mice after treatment with the same amount of i.p. cyclophosphamide [68]. Although we observed CD45.1^+^ cells of the myeloid lineage (CD45.1^+^ CD11b^+^) in the blood, spleen and tumor, reconstituted cells, in contrast to their host counterparts, did not give rise to relevant amounts of progeny with the phenotype of PMN- or M-MDSC or displayed a similar behavior in frequency kinetics over time. Thus, we conclude that CD45.1^+^ MDSC do not considerably contribute to an immunosuppressive tumor micromilieu in mice with established D5 melanoma.

To enhance priming of tumor-specific T cells and anti-tumor effects of cytotoxic T cells, we aimed to deplete CD11b^+^ Gr-1^+^ cells. Anti-Gr-1 mAb (clone: RB6-8C5) has already successfully been used to eliminate MDSC in tumor, spleen, peripheral blood and bone marrow of tumor-bearing and control mice [25,27,43,44,45]. Thus, proof of principle for the efficacy of the anti-Gr-1 antibody and its therapeutic effect slowing down the growth of malignant tumors has already been brought forward. Srivastava et al., for example, used 200 µg anti-Gr-1 mAb (RB6-8C5) every other day for a total amount of 4 weeks in a model of 3LL-lung carcinoma, starting one week after tumor inoculation. They observed a reduction of CD11b^+^ Gr-1^+^ in blood, spleen, bone marrow and tumor and a significant reduction in tumor volume and weight [27]. Using a similar approach in mice carrying 3LL-tumors, Zhang et al. were able to significantly reduce tumor-infiltrating MDSC, slowing down tumor growth and improving survival of the animals, using repeated i.p. administrations of 250 µg RB6-8C5, every 3 days starting 2 weeks after tumor inoculation [44].

In our experiments, we administered 230 µg anti-Gr-1 mAb (RB6-8C5) or isotope control via intraperitoneal injection every other day, starting with the day of cyclophosphamide administration and continuing until the animals were euthanized, thus ensuring that the period of RB6-8C5 administration would cover the days with maximum MDSC frequencies. To maintain comparability, dose and time intervals between the single doses of antibody were set in accordance with previous reports [26,44]. However, assessing the depletion status of MDSC is not trivial since fluorochrome labelled Ly6G antibodies (clone 1A8), which we and others used to distinguish between PMN- and M-MDSC, do not bind due to the same binding site, when anti-Gr-1 antibodies were administered previously [25,26,28]. Thus, we performed co-staining and quantification of RB6-8C5-bound cells using a secondary antibody-approach with fluorochrome-labeled antibodies directed against goat-IgG heavy chains of the anti-Gr-1 antibodies [25,26,28,48]. Complete absence of MDSC after conventional and secondary antibody staining indicated that MDSC were reliably depleted. Absence of MDSC regarding the conventional staining, but detection of secondary Ab positive cells in the same gate or neighboring areas, indicated the persistence of MDSC. In some cases, we observed (a) secondary Ab bound cells negatively stained for Ly6G, (b) a cell cloud with a positive and negative portion regarding Ly6G-staining, or (c) secondary Ab bound cells within the PMN-gate. The fact that cells of all three cases retained the same Ly6C intensity suggests that in every case PMN-MDSC with different degrees of Gr-1 epitope saturation with RB6-8C5 were visualized. Hence, the absence of PMN-MDSC in their designated gate was not necessarily accompanied with complete absence or depletion of PMN-MDSC.

With this in mind, we observed complete disappearance of PMN-MDSC in the peripheral blood, 2 and 4 days after treatment initiation, as there were no secondary antibody-bound cells in the PMN-gate. However, starting with day 4, RB6-8C5-bound cells emerged in the Ly6C^mid^-Ly6G^−^ gate, indicative for PMN-MDSC completely covered with RB6-8C5 mAb and thus potentially mimicking successful depletion (Appendix A). In contrast, the frequencies of M-MDSC appeared to be reduced but were not completely depleted. In the timespan, when maximum frequencies of MDSC were to be expected following LRAST treatment (day 7 and day 10), frequencies of PMN-MDSC appeared to be significantly reduced, but, at the same time, a major portion of cells in the PMN-gate showed secondary Ab binding. Since anti-Gr-1 antibodies (RB6-8C5) were shown to persist on the cell surface of MDSC for up to 4 days and might retain suppressive activity [25,26], the persisting portion of RB6-8C5-bound cells might represent an obstacle to therapy due to preserved immunosuppressive properties. Therefore, we aimed at investigating whether the observed reduction of PMN-MDSC, despite the occurrence of RB6-8C5-bound cells, would be sufficient to improve a tumor-specific T cell response and display a measurable therapeutic effect.

IFN-γ secretion is often used as a marker for the cytotoxic properties of T cells, including anti-tumor reactivity [69,70,71,72]. Van den Engel et al. have already demonstrated an increase in IFN-γ secretion from TVDLN after LRAST [7]. Here, we hypothesized that tumor-specific T cells in TVDLN from mice in the group with LRAST + RB6-8C5 treatment would exert improved IFN-γ producing capability due to better T cell-priming after MDSC depletion. TVDLN from mice treated with LRAST alone already presented an increased tumor-specific IFN-γ production and a delay in tumor outgrow compared to untreated mice. We could show that RB6-8C5 administration in addition to LRAST could further increase the IFN-γ-secretion from TVDLN, although significant results could not be obtained due to the variability of results per mouse. In accordance with this, tumor growth in mice treated with LRAST + RB6-8C5 appeared significantly reduced in the initial treatment phase up to 13 days after tumor inoculation, including the point in time with maximum IFN-γ secretion and covering the timespan of successful MDSC depletion in our experiments—at least regarding PMN-MDSC. Since PMN-MDSC are known to mainly target T cell priming accounting for tumor-specific T cell tolerance [24], the improvement in tumor-specific INF-γ secretion from TVDLN and the delay of tumor-growth, which is chronologically fitting to the time-span of successful reduction of PMN-MDSC, implies that the observed effects are attributable to the administration of MDSC-depleting antibodies. In contrast to Srivastava et al. and Zhang et al. [27,44], who gained good long-term results due to a presumably successful long-term depletion of MDSC in their tumor models, our results clearly demonstrate the recurrence of MDSC, especially the PMN-subset after repetitive administration of anti-Gr-1 antibodies. The presence of RB6-8C5-bound cells, as indicated by secondary Ab binding, initially increased over time. After approximately 4 weeks of repetitive RB6-8C5-administration, binding of depleting antibodies could not be observed anymore and MDSC depletion appeared insufficient. The comparability of our data to pre-existing literature on MDSC depletion using RB6-8C5 is impaired due to a pervasive pre-treatment, which even bears the risk to be a MDSC driving stimulus itself. We assume that most likely side effects of the components of LRAST work in synergy and oppose MDSC eradication. Hence, despite its positive immunomodulating features, low-dose cyclophosphamide is known to increase levels of various cytokines (GM-CSF, IL-1b, IL-5, IL-10, IFN-γ, TNF-α), which can contribute to the expansion and activation of MDSC [73]. GM-CSF, a major component within LRAST, in turn is a driving force in MDSC recruitment. It is not only found manifold as an adjuvant in immunotherapeutic regimes, but also produced by many human and murine tumor cell lines [74]. GM-CSF has been shown to recruit MDSC into secondary lymphoid tissues with a consecutively impaired function of tumor-specific CD8^+^ T cells [75]. In experiments with irradiated GM-CSF producing B78H1-GM cells, a cell line derived from the B16 melanoma, Serafini et al. proposed a cut-off concentration at 1500 ng/10^6^ cells/24 h, which—if exceeded—was associated with a suppression of antigen-specific T cell response [76]. Irradiated D5G6 cells used in this work showed an in vitro GM-CSF production of 154 ng/10^6^ cells/24 h with an ascending tendency over 6 days (Appendix A). Although the GM-CSF concentrations observed in our experiments are below the proposed cut-off, a positive contribution to MDSC recruitment by GM-CSF cannot be ruled out, especially considering the potentially synergistic effects when combined with cyclophosphamide. Furthermore, even the anti-Gr-1-depleting antibodies themselves may act as a driver for MDSC expansion and thereby stand in the way of their own therapeutic purpose. Single application of RB6-8C5 was shown to be accompanied by enlarged spleens with increased cell numbers 9 days after antibody injection [43], observations which we also could obtain in our experiments. An increase in numbers of PMN- and M-MDSC was attributed to a pronounced proliferative stimulus on early myeloid precursors due to depletion [43]. More to the point, following repetitive administration of RB6-8C5 every other day, Ribechini et al. found an induction of STAT1, STAT3 and STAT5. STAT3 in particular acts as an inducer of myeloid cell lineages and thereby may promote MDSC differentiation and activation [26].

Overall, the increasing frequencies of PMN-MDSC despite administration of anti-Gr-1 mAb, and an increasing number of cells with cell-bound RB6-8C5, indicate that antibody administration becomes insufficient in keeping up with the reproduction/regeneration of MDSC, especially the PMN-subset. The abrogated depleting efficacy of long-term use of RB6-8C5 together with the fact that, in the long run, no RB6-8C5 bound cells were detectable in the gates of both MDSC-subsets, implies the existence of a neutralizing mechanism (e.g., neutralizing self-antibodies) directed against RB6-8C5 antibodies.

Depending on the tissue localization the ratio of MDSC subpopulations and their suppressive activity vary, as does their susceptibility to RB6-8C5 [13]. Overall, we observed a differing behavior of the M-MDSC subset compared to their PMN counterparts in response to MDSC depletion. For a period of 4 weeks, repetitive RB6-8C5 administration did not—apart from an initial reduction in peripheral blood—have any significant depleting effect on M-MDSC in the peripheral blood of tumor bearing mice. In previous reports particularly tumor-localized M-MDSC were shown to be resistant to depletion with anti-Gr-1 mAb and the frequency of Ly6C^high^ cells was not altered 48 h after a single administration of 250 µg RB6-8C5 [26,77]. In our experiments, frequencies of tumor-infiltrating PMN- and M-MDSC showed no significant difference after LRAST and repetitive administration of RB6-8C5, compared to mice treated with LRAST alone (Appendix A). Apart from that, varying levels of Gr-1-expression on both MDSC subsets, as anti-Gr-1 mAbs bind to both Ly6G and Ly6C, might affect binding capacity of RB6-8C5 and therefore influence depleting results [78]. Since the tumor-specific milieu as well as cytokines exerted by various leukocytes affect MDSC generation, activation and distribution, we expect the increased cytokine levels after lymphopenia induction with cyclophosphamide including GM-CSF and the GM-CSF released by the whole-cell vaccine to have relevant influence on M-MDSC kinetics. Lesokhin et al. reported that chronic GM-CSF exposure in a B16-GM-melanoma model leads to increased expansion of CCR2^+^ monocytic MDSC and accumulation at the tumor site. This prevented adoptively transferred activated CD8^+^ T-cells from entering the tumor site [79]. Thus, we assume that in our experiments, the variability in M-MDSC frequencies by day 15 and 24 after LRAST pretreatment (w/o RB6-8C5) results from a GM-CSF-driven expansion and redistribution, e.g., into tumor tissue. Still, given the initial occurrence of RB6-8C5 bound cells in the M-MDSC gate and a short-term reduction in numbers of M-MDSC after treatment initiation with LRAST + RB6-8C5, we cannot assume a general resistance of M-MDSC to anti-Gr-1 mAbs. However, M-MDSC frequency kinetics appeared to be independent of anti-Gr-1 Ab administration. This has to be evaluated in future experiments.

Apart from the MDSC-depleting properties of anti-Gr-1 antibodies, a secondary focus was to elucidate the effect of RB6-8C5 on CD8^+^ cells and CD8^+^ memory T cell subsets (T_cm_ and T_em_) in mice after LRAST treatment. Both the anti-Gr-1 antibodies as well as lymphopenia induced by cyclophosphamide provably have an impact on CD8^+^ T cells and their memory subsets [11,47,48], but—to the best of our knowledge—the combined effect has not been investigated thus far. In line with previous work by Matsuzaki et al., we confirmed the expression of Gr-1 on memory CD8^+^, but not CD4^+^ T cells with FACS analysis (data not shown) and observed a depletion of CD8^+^ memory T cell subsets following RB6-8C5 administration [48]. In accordance to results from Salem et al., the relative cell frequency kinetics of CD8^+^ cells were obviously influenced by lymphopenia/lymphopenia driven homeostatic proliferation [68], but no significant impact of MDSC depletion on the population of CD8^+^ cells could be observed. However, looking at the CD8^+^ memory subsets (T_cm_ and T_em_), reduced staining capability with fluorochrome-conjugated mAb directed towards Gr-1 and reduced overall frequencies indicated that not only targeting of memory T cells by anti-Gr-1 mAb occurred, but a portion of both memory T cell subsets was depleted following RB6-8C5 administration, independent from pretreatment with LRAST. Nevertheless, despite continuous injections with anti-Gr-1 mAb, we observed that frequencies of T_em_, more than T_cm_, in the peripheral blood strongly increased following LRAST, reflecting in a T_em_:T_cm_ ratio of 3.5:1 by day 7, compared to a ratio of 1.2:1 in the control group. T_em_ in mice with RB6-8C5 monotherapy and LR + RB6-8C5 treatment (no vaccination) remained reduced in the same period of time. These results are in line with observations from Ma and coworkers. They found that the exposure to a tumor vaccine during homeostatic recovery after induction of lymphopenia resulted in strong expansion of CD4^+^ and CD8^+^ CD44^high^ CD62L^low^ effector memory T cells, accompanied by a pronounced tumor-specific IFN-γ production and better tumor reactivity [11]. In our experiments, the proliferative drive seems to exceed the depleting properties of the anti-Gr-1 antibodies exerted upon the Gr-1 expressing CD8^+^ memory T cells, which are capable of producing significant amounts of IFN-γ in a tumor-specific manner [11,48]. Nevertheless, it remains to be evaluated whether the observed increase in IFN-γ secretion from TVDLN is attributable to the increased occurrence of T_em_, since these cells lack the ability to home to lymphatic tissue [37]. Also, compared to CD8^+^ T_em,_ CD8^+^ T_cm_ are referred to as the cell population with a more pronounced role in antitumor immunity. Downregulation of CD62L enables T_em_ to quickly migrate to peripheral tissues and exert effector functions upon antigen encounter [39]. Thus, T_em_ might successfully invade tumor tissue and kill transformed cells. We therefore hypothesize that the increased frequencies of T_em_ cells might be reflected in the initial delay of tumor growth of mice treated with LRAST + RB6-8C5. Nevertheless, the influence of RB6-8C5 on the CD8^+^ memory T cell compartment might limit the anti-tumor efficacy of LRAST + RB6-8C5 treatment.

## 5. Conclusions

Overall, within the wide field of immunotherapy, cancer combination therapies continue to be promising. With LRAST, a therapy combining lymphodepletion with active-specific tumor cell vaccination, we could confirm pre-existing results showing a notable delay in melanoma growth. By adding the use of monoclonal-depleting antibodies (anti-Gr-1 mAb), our data highlight the significance of MDSC as major tumor-promoting cells, as depletion of MDSC further delayed tumor progression and improved the tumor-specific IFN-γ response. Nevertheless, the positive effects of MDSC depletion in addition to LRAST were mainly restricted to the PMN subset and the first weeks after treatment initiation. The recurrence of PMN-MDSC and the impact of anti-Gr-1-depleting antibodies on the memory T cell compartment might be responsible for the failure in long-term tumor reduction. Overall, we could show that reduction of the frequencies of MDSC in animals with established melanoma has beneficial effects.

## 6. Limitations

Leukocyte dynamics as assessed by flow cytometry at different time points in spleen and blood, as well as tumor, provide profound insights into immunological response to therapy. Nevertheless, functional analysis of both MDSC subtypes in different organs and at different time points would help to better understand their suppressive capacities and evaluate their mechanisms of suppression. A detailed description of leukocyte populations, their functional status and interplay in the tumor micromilieu would also provide guidance to further improve LRAST therapy with MDSC depletion and should be performed in future experiments.

## Figures and Tables

**Figure 1 vaccines-10-00560-f001:**
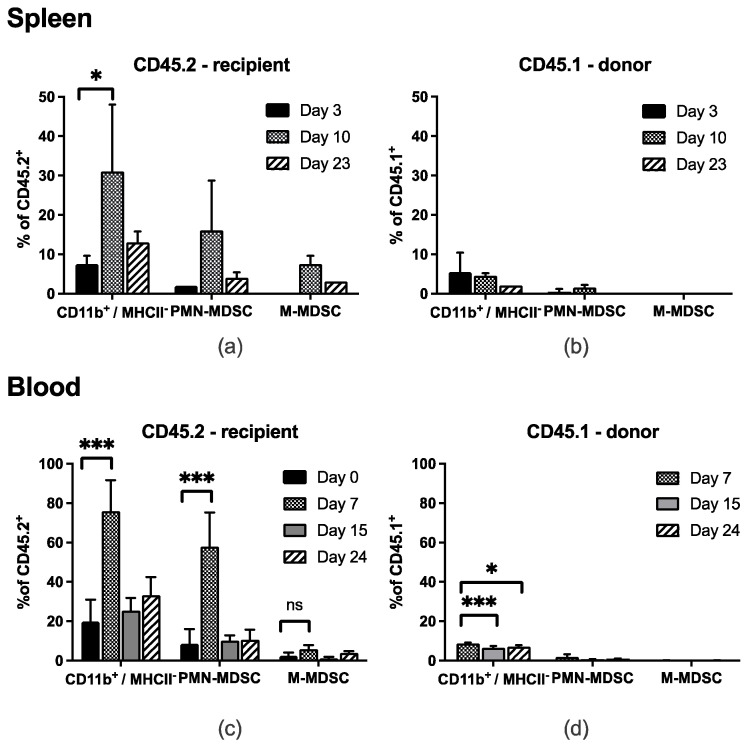
**Myeloid cells from spleen and blood at different time points after LRAST-treatment**. C57BL/6 mice were exposed to 5 × 10^4^ D5-cells by s.c. injection and treated according to the LRAST scheme with cyclophosphamide (L) and active-specific vaccination (AST). CD11b^+^ MHC-II^−^ cells, PMN-MDSC (CD11b^+^ Ly6G^+^ Ly6C^+^) and M-MDSC (CD11b^+^ Ly6G^−^ Ly6C^high^) from spleen (**a**,**b**, *n* = 2) and blood (**c**,**d**, *n* = 3) were determined on days 3, 10 and 23 (spleen) and 0, 7, 15 and 24 (blood, except for CD45.1^+^ donor cells, which were not in situ of the recipient mice on day 0). Frequencies of the different myeloid cell types are depicted as percentages of CD45^+^ leucocytes and separated into CD45.2^+^ (recipient mouse, **a**,**c**) and CD45.1^+^ (donor mouse, **b**,**d**) cells, respectively. Data are means with SD, and two-way ANOVA was performed. Symbols indicate: * *p* < 0.05; *** *p* < 0.001.

**Figure 2 vaccines-10-00560-f002:**
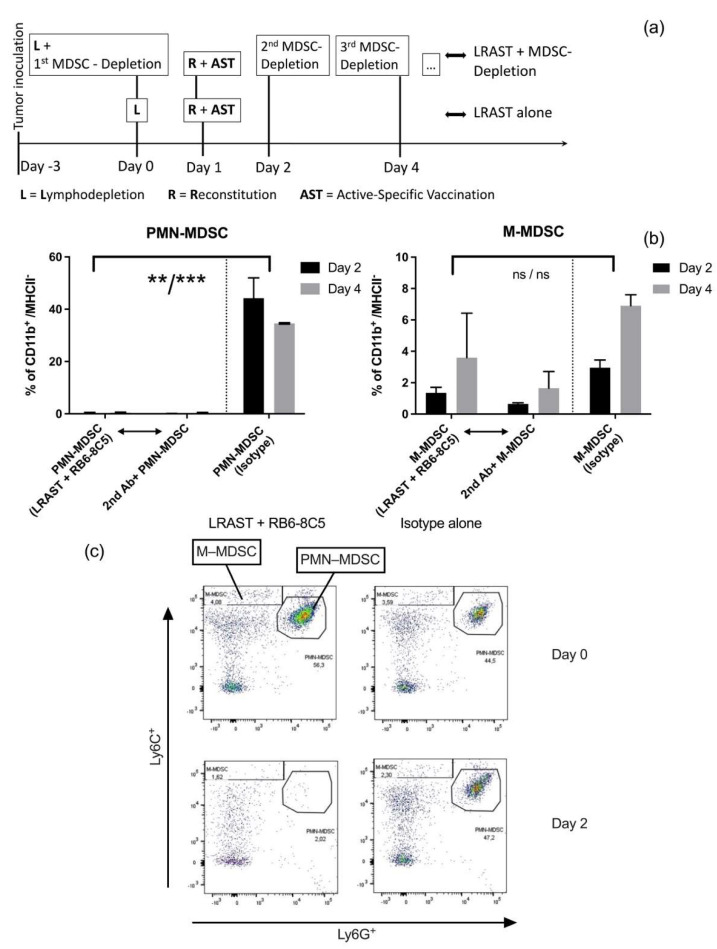
**Successful elimination of PMN–MDSC in the initial treatment phase of LRAST + RB6-8C5**. (**a**) LRAST treatment scheme. D5 tumor cells (5 × 10^4^) were injected s.c. into C57BL6 mice 3 days prior to LRAST treatment. One day following lymphopenia induction with cyclophosphamide (200 mg/kg, i.p.), C57BL/6 (CD45.2^+^) mice were reconstituted i.v. with 20 × 10^6^ splenocytes from naïve C57BL/6 (CD45.1^+^) mice and vaccinated s.c. with 10 × 10^6^ irradiated D5G6 cells. Beginning on the day of lymphodepletion, in groups with MDSC depletion, mice received 230 µg anti-Gr-1 mAb (clone: RB6-8C5) i.p. every other day until mice were euthanized. *n* = 2 per group. (**b**) Plots depict mean and standard deviation visualizing depletion results assessed by flow cytometry for PMN-MDSC and M-MDSC on days 2 and 4 from the blood of mice treated with LRAST + RB6-8C5 compared to control (Isotype alone), respectively. Results in the middle of both plots represent % of CD11b^+^/MHCII^−^ cells with RB6-8C5-binding as indicated by secondary antibody staining (“2nd Ab+”). Symbols indicate: ** *p* < 0.01; *** *p* < 0.001; ns *p* > 0.05. Student’s *t* test was performed. (**c**) Representative dot plots for (**b**). Dot plots demonstrate depletion of PMN- and M-MDSC on day 2 (=48 h) after treatment initiation with LRAST + RB6-8C5-treated mice compared to mice receiving Isotype control Ab without any further treatment.

**Figure 3 vaccines-10-00560-f003:**
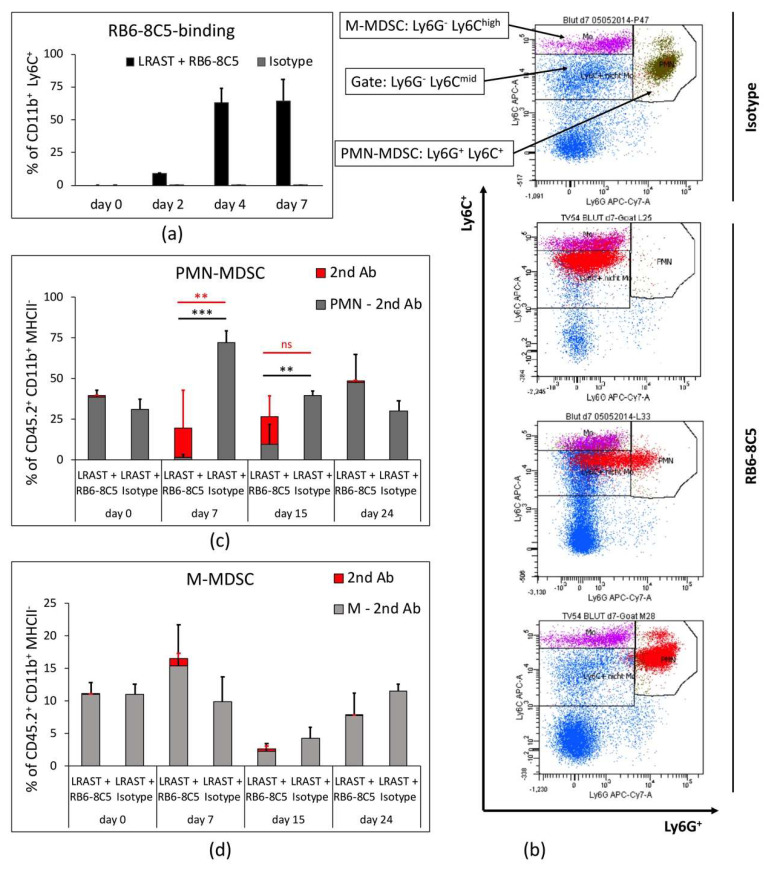
**Long–term results of MDSC depletion and appearance behavior of PMN–MDSC throughout the gates**. (**a**) As assessed by flow cytometry columns represent % of CD11b^+^ Ly6C^+^ cells with anti-Gr-1 mAb (RB6-8C5) bound to their cell surface in the blood of mice treated with LRAST + RB6-8C5 compared to mice treated with Isotype Ab alone. RB6-8C5-binding is as indicated by secondary antibody staining. FACS analysis was performed on days 0 (before first RB6-8C5 administration), and days 2, 4, and 7. *n* = 2 per group. (**b**) Representative Ly6G/Ly6C dot-plots with gates for PMN-, M-MDSC, and Ly6C^mid^/Ly6G^−^ cells, illustrating appearance of PMN-MDSC when exposed to depleting antibodies (RB6-8C5) on day 7. Red dots represent RB6-8C5-bound cells, as indicated by secondary antibody staining. (**c**,**d**) Long-term depletion results for PMN-, and M-MDSC, depicted as % of CD45.2^+^ CD11b^+^ MHCII^−^ cells on days 0 (before first RB6-8C5 administration), and days 7, 15, and 24 after lymphodepletion. Grey columns represent the amount of PMN- or M-MDSC net of cells with RB6-8C5 surface binding within the same (PMN- or M-MDSC-) gate, respectively. Red columns represent the portion of secondary antibody bound cells (“2nd Ab”) within the PMN- or M-MDSC gate, respectively. *n* = 3–4 per group. (**a**,**c**,**d**) Data are means with SD. Two-way ANOVA was performed. Symbols indicate: ** *p* < 0.01; *** *p* < 0.001; ns *p* > 0.05.

**Figure 4 vaccines-10-00560-f004:**
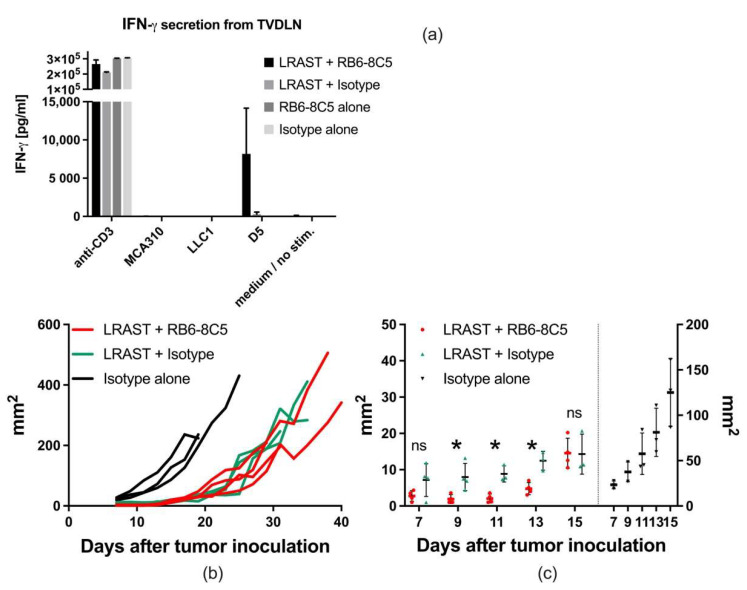
**Cytokine response and effect on tumor growth after combining LRAST with anti-Gr-1 antibody treatment**. (**a**) C57BL/6-mice were subcutaneously injected with 20 × 10^4^ D5-cells (5 × 10^4^ at each flank nearby the proximal extremities) and treated according to LRAST w/o MDSC depletion with anti-Gr-1 mAb (clone RB6-8C5) (*n* = 2–4 per group). Mice were euthanized on day 9 after lymphodepletion, and inguinal as well as axillary lymphatic nodules were harvested. T cells from TVDLN were polyclonally stimulated with anti-CD3 mAb and expanded with IL-2. To assess tumor-specific IFN-γ release, cells were incubated with autologous tumor cells of the D5-lineage or control tumor cell-lines (MCA310 and LLC1). Wells coated with anti-CD3 mAb served as positive controls, and wells with native RPMI cell culture medium represented negative controls. Bars represent means with SEM. (**b**) Subcutaneous tumor growth of mice treated with LRAST alone (green), LRAST combined with RB6-8C5 (red), or isotype control (black) (*n* = 4 for both LRAST groups, *n* = 3 for isotype control group). Maximum and minimum diameter of the tumors were determined using caliper and the multiplication product (in mm^2^) depicted against time (in days). Each line represents one mouse of the respective treatment group. (**c**) Comparison of mice treated with LRAST alone versus mice treated with a combination of LRAST and RB6-8C5, focusing on the initial treatment phase (days 7–15). Dots represent values of single mice. Error bars indicate means with SD. Student’s *t* test was performed. Symbols indicate: * *p* < 0.05; ns *p* > 0.05.

**Figure 5 vaccines-10-00560-f005:**
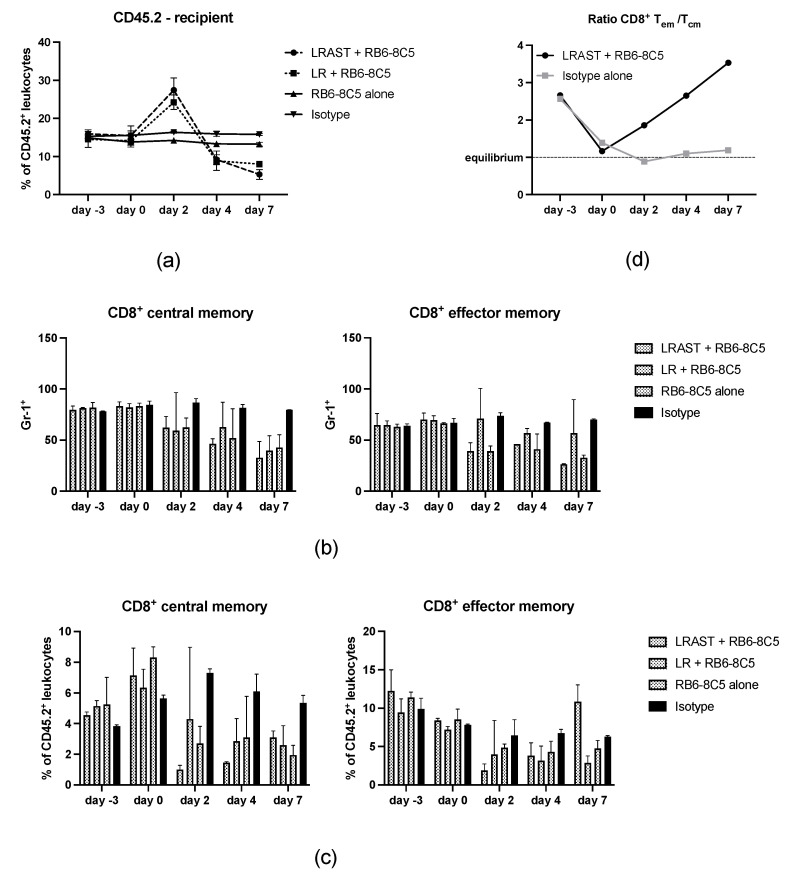
**Effect of RB6-8C5 administration on CD8^+^ memory T cells**. (**a**) Frequency of CD3^+^ CD8^+^ cells in % of CD45.2^+^ leukocytes (in the following referred to as CD8^+^ cells), assessed by flow cytometry on days -3, 0, 2, 4, and 7 from the blood of mice in different treatment groups (as indicated in the picture). (**b**) Frequency of Gr-1^+^ CD8^+^ central (left) and effector (right) memory T cells from the same animals as in (**a**). (**c**) CD8^+^ central (left) and effector (right) memory T cells in % of CD45.2^+^ leucocytes form the same animals as in (**a**). (**d**) Ratio of CD8^+^ effector and central memory T cells (T_em_/T_cm_) calculated from results depicted in (c). A ratio equal to 1 represents an equilibrium state with identical amounts of both memory T cell subsets. (**a**–**d**) *n* = 2 per group. Data are means with SD.

## Data Availability

The data presented in this study are available from the corresponding author on reasonable request.

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
