# Peer review of "Anti-Gr-1 Antibody Provides Short-Term Depletion of MDSC in Lymphodepleted Mice with Active-Specific Melanoma Therapy"

_vaccines, 2022, doi:10.3390/vaccines10040560_

Round 1

Reviewer 1 Report

An interesting original article evaluating a possible benefit of Myeloid-derived suppressor cells depletion using anti-Gr-1 antibodies (Ab) in combination with lymphodepletion, reconstitution and active-specific tumor cell vaccination. Only minor queries:

Line 36 you should add: "Given the high risk of disease recurrence in thicker, nodular or mucosal forms, new systemic treatments are necessary for the management of this condition." and cite an article such as doi: 10.3390/medicina57040359. 

In the statistical analysis section did you use two versions of Graphpad? I did not understand...

Good Luck!

Author Response

Response to Reviewer 1 Comments

Dear Reviewer,

thank you very much for your revision of our manuscript “Anti-Gr-1 antibody provides short-term depletion of MDSC in lymphodepleted mice with active-specific melanoma therapy”.

Point 1:

Line 36 you should add: "Given the high risk of disease recurrence in thicker, nodular or mucosal forms, new systemic treatments are necessary for the management of this condition." and cite an article such as doi: 10.3390/medicina57040359. 

Response 1: The comment to mention disease recurrence of melanoma in thicker, nodular or mucosal forms is very important. I felt it would be better placed in the discussion and added the information in line 431-433.

Point 2:

In the statistical analysis section did you use two versions of Graphpad? I did not understand...

Response 2:

We started with Version 7 and switched to Version 8. In the end all graphs and statistics were run with version 8 before submission, so I deleted “Version 7”.

Best regards,

Peter Rose

Reviewer 2 Report

In the current manuscript the Authors investigated the possible benefit of MDSC depletion using anti-Gr-1 antibodies (Ab) in combination with an active vaccination protocol involving tumor-cell vaccination with irradiated autologous mGM-CSF-secreting melanoma cells in a preclinical melanoma model.

The study does not address specific novel concepts and it is lacking of many experimental determinations, for example an analysis of the tumor microenvironment:

Lymphodepletion followed by whole cell vaccine combined with GM-CSF was already described to be able to enhance the anti-tumor immune response in preclinical tumor models. Moreover, different efforts have been made so far to ablate or modulate MDSC including the usage of anti-Gr-1 or anti-Ly6G monoclonal antibodies (mAb) as here performed. Thus, the remaining focus of this paper rely on addressing the therapeutic efficacy of such combination. Knowing that, besides the lymphodepleting effect, also GM-CSF has been described to exacerbate myelopoiesis the Authors should also provide:

  • Analysis of MDSC functional activity (differentiation/activation), besides frequency in this regimen.
  • Evidence of intra-tumoral and bone-marrow changes related to both MDSC and CD8+ T cells
  • Address regulatory T cell frequency.
  • Address suppressive activity of MDSC through dedicated CFSE-based T cell suppression assays
  • IHC analysis on tumor tissues.
  • In all Figure Captions pleas indicate number of mice per group, statistic test used and if bars indicate SD or SEM.
  • Effects on M-MDSC seems to be aspecific? Fig 3d, please comment. Please discuss more abount MDSC kinetics
  • 4b Please comment why any reduction in tumor volume was recorded.

Given the conclusion of the manuscript, that in such therapeutic regimens anti-MDSC abs failed to provide long-term MDSC depletion I would suggest make the title itself more focus

Minor points

Pag 6 Fig 2a Please ameliorate definition of schedule treatment

Pag 6 Fig 2c Pag 6 Fig 2c Please indicate name on the axis

Pag 6 Fig 2d M-MDSC (isotype) how has been stratify this group?

Pag 6 Fig 3b Please ameliorate Panel, seem time point are missing in the figure

Pag 9 Line 338 “tumor-cell lines (LLC1, mGC8, MCA310)” why mGC8 was cited?

Pag 11 Fig 5b, please include statistics, number animals/groups etc etc…

Author Response

Dear Reviewer 2.

For my response to your comments please see the attached word file.

Best regards,

Peter Rose

Reviewer 3 Report

The article “Anti-Gr-1 antibody provides short-term depletion of MDSC in 2 lymphodepleted mice with active-specific melanoma therapy” discusses an important problem in the treatment of cancer - melanoma. This is a very important issue as we still do not have a satisfactory method of treating melanoma, especially in advanced stages. I think this topic is original and useful. The conclusions are in line with the evidence and arguments presented in the study The references presented in the article are correct. Alternatively, you can supplement the references with newer articles, because the newest one is from 2018 and there is no information from later years here. The topic is very interesting but would require application of this therapy in the in vitro culture of human-derived cancer cells in the future.

I have the biggest objection to the figures:

Figure 2a - poor quality, this needs to be improved.

Figure 2b - an unreadable graph, poorly illustrating the data.

Figure 4 - poor quality, this needs to be improved.

After improving the quality of the figures, I think that the article is suitable for publication.

Author Response

Response to Reviewer 3 Comments

Dear Reviewer,

thank you very much for your revision of our manuscript “Anti-Gr-1 antibody provides short-term depletion of MDSC in lymphodepleted mice with active-specific melanoma therapy”.

Point 1:

The references presented in the article are correct. Alternatively, you can supplement the references with newer articles, because the newest one is from 2018 and there is no information from later years here. 

Response 1: I added latest articles regarding MDSC in tumorimmunology.

  • Please see reference list: [1-7]

Objections to the figures:

Point 1 + 2:

  • Figure 2a - poor quality, this needs to be improved.
  • Figure 2b - an unreadable graph, poorly illustrating the data.

Response 1 + 2:

Fig 2a was improve by re-arranging the elements of the timeline. Additionally, I changed the type of graph for Fig 2b to improve understandability. All was saved with higher quality / resolution.

Point 3:

Figure 4 - poor quality, this needs to be improved.

Response 3:

Fig.4 was reformatted and saved with higher quality.

Best regards,

Peter Rose

References:

  1. Law, A.M.K.; Valdes-Mora, F.; Gallego-Ortega, D. Myeloid-Derived Suppressor Cells as a Therapeutic Target for Cancer. Cells 2020, 9, doi:10.3390/cells9030561.
  2. Veglia, F.; Sanseviero, E.; Gabrilovich, D.I. Myeloid-derived suppressor cells in the era of increasing myeloid cell diversity. Nat Rev Immunol 2021, 21, 485-498, doi:10.1038/s41577-020-00490-y.
  3. Yang, Y.; Li, C.; Liu, T.; Dai, X.; Bazhin, A.V. Myeloid-Derived Suppressor Cells in Tumors: From Mechanisms to Antigen Specificity and Microenvironmental Regulation. Front Immunol 2020, 11, 1371, doi:10.3389/fimmu.2020.01371.
  4. Hao, Z.; Li, R.; Wang, Y.; Li, S.; Hong, Z.; Han, Z. Landscape of Myeloid-derived Suppressor Cell in Tumor Immunotherapy. Biomark Res 2021, 9, 77, doi:10.1186/s40364-021-00333-5.
  5. Tengesdal, I.W.; Menon, D.R.; Osborne, D.G.; Neff, C.P.; Powers, N.E.; Gamboni, F.; Mauro, A.G.; D'Alessandro, A.; Stefanoni, D.; Henen, M.A.; et al. Targeting tumor-derived NLRP3 reduces melanoma progression by limiting MDSCs expansion. Proc Natl Acad Sci U S A 2021, 118, doi:10.1073/pnas.2000915118.
  6. Hassel, J.C. Checkpoint blocker induced autoimmunity as an indicator for tumour efficacy in melanoma. Br J Cancer 2022, 126, 163-164, doi:10.1038/s41416-021-01390-1.
  7. Lombardo, N.; Della Corte, M.; Pelaia, C.; Piazzetta, G.; Lobello, N.; Del Duca, E.; Bennardo, L.; Nistico, S.P. Primary Mucosal Melanoma Presenting with a Unilateral Nasal Obstruction of the Left Inferior Turbinate. Medicina (Kaunas) 2021, 57, doi:10.3390/medicina57040359.

Round 2

Reviewer 2 Report

Given the Authors'reply to me there are still many aspects / replicates to be filled that I do not feel the manuscript is acceptable in its present form.

Author Response

Dear Reviewer, thank you for your comment.